# CodePlot-CoT: Mathematical Visual Reasoning by Thinking with Code-Driven Images

Figure 1: A comparison of mathematical reasoning benchmarks and the methods on the visual reasoning problem. (1) illustrates that unlike existing benchmarks that rely on textual reasoning, Math-VR requires deep visual reasoning to resolve the math problems. (2) shows that on a visually ambiguous problem from Math-VR, both text-only and unified multimodal models fail. Our method, CodePlot-CoT, succeeds by programmatically generating the figure to uncover its true geometric properties, thus arriving at the correct solution.

## Abstract

Recent advances in Large Language Models (LLMs) and Vision Language Models (VLMs) have shown significant progress in mathematical reasoning, yet they still face a critical bottleneck with problems requiring visual assistance, such as drawing auxiliary lines or plotting functions to solve the problems. Most LLMs and VLMs are constrained to text-only reasoning chains, while multimodal unified models that can generate interleaved text and images lack the necessary precision and controllability for such tasks. To address this, we propose **CodePlot-CoT**, a code-driven Chain-of-Thought paradigm for "thinking with images" in mathematics. Our approach leverages the VLM to generate text reasoning as well as executable plotting code, which is then rendered into images as "visual thought", to solve mathematical problems. To achieve this, we first construct **Math-VR**, the first large-scale, bilingual dataset and benchmark for Mathematics problems with Visual Reasoning, comprising 178K samples. Second, to create high-quality training data, we develop a state-of-the-art image-to-code converter specialized for parsing complex mathematical figures into codes. Finally, using these training data, we train the CodePlot-CoT model for solving mathematical problems. Experimental results show that our model achieves up to **21%** increase over base model on our new benchmark, fully validating the efficacy of our proposed code-driven reasoning paradigm. Our work opens a new direction for multimodal mathematical reasoning and provides the community with the first large-scale dataset, comprehensive benchmark, and strong approach for such problems.

> *"Algebra is but written geometry and geometry is but figured algebra."*
>
> – Sophie Germain

# 1 INTRODUCTION

Human cognition is inherently multimodal, leveraging visual graphs, diagrams, and sketches to facilitate complex reasoning. This is particularly evident in mathematics, where such visual aids—from drawing auxiliary lines in geometric proofs to plotting functions—are essential for rendering abstract relationships concrete and making the reasoning process more intuitive (Hu et al., 2024). While recent advances in Vision Language Models (VLMs) have shown strong performance in mathematical reasoning (Gao et al., 2023a; Shi et al., 2024; Zhang et al., 2024b), they typically reply on text-only reasoning chains. This becomes a major limitation for problems that require visual reasoning, where humans would simply sketch diagrams or add auxiliary lines to facilitate reasoning. Such deficiency in multimodal reasoning leads to redundant and even incorrect text-only reasoning in mathematical problem solving (shown in Figure 1).

Recent efforts in general-domain visual understanding have explored the paradigm of Visual Chain-of-Thought (Visual CoT) (Shao et al., 2024), attempting to realize it by directly generating and manipulating images (Li et al., 2025b; Chen et al., 2025; Li et al., 2025a). However, this paradigm breaks down in the context of mathematics problems that demand high precision, where naive image generation is insufficient. Even state-of-the-art unified models struggle to execute precise operations in mathematics, such as constructing auxiliary lines that satisfy strict geometric constraints.

The fundamental challenge in direct image generation and manipulation arises from the inherent difficulty in modeling the high-dimensional distribution of natural images, which contain complex textures and high-frequency details. However, mathematical visual aids differ significantly from general image generation tasks: the critical aspect is not the pixel-level details or textures, but rather the precise representation of key structured geometric properties such as shapes, lengths, positions, and angular relationships. This distinction suggests that the essential information in mathematical figures can be better captured by structured representations rather than pixel-level encodings. Therefore, we introduce programmatic code as the optimal representation for mathematical visual reasoning. As an inherently textual and structured representation format, code aligns seamlessly with language models (Surís et al., 2023; Wang et al., 2025a; 2023), enabling straightforward generation without introducing complex distributional modeling (*e.g.*, diffusion models).

In this work, we propose a new paradigm that enables VLMs to engage in visual reasoning through code generation. Instead of directly generating images with VLMs, which typically leads to degraded quality and precision in math plots, our approach guides the model to output executable plotting codes which are rendered into images as intermediate "visual thoughts". Once executed, the generated code produces images that can be input back into the VLM reasoning sequence.

Implementing this paradigm requires addressing two key challenges. First, there is a lack of structured dataset and benchmark for mathematical problems that demand visual reasoning, as existing works focus mainly on interpreting given figures rather than reasoning with visual images during problem solving (Lu et al., 2023; Zhang et al., 2024a). Therefore, we construct Math-VR, a large-scale bilingual dataset and benchmark comprising 173K training, 5K testing mathematical problems with visual reasoning solutions. We then benchmark existing SOTA models, thereby establishing strong baselines and highlighting the difficulty of this new task. Second, training models to output code as representations of visual thought requires a bidirectional mapping between code and images. We tackle this by developing MatplotCode, a high-fidelity image-to-code converter, which we leverage to construct code-driven CoT for training. The curated data then serves as the foundation for training CodePlot-CoT, a model specialized for code-driven visual reasoning. Our experiments show that it achieves a up to 21% increase over base model, validating the efficacy of our approach.

The main contributions of our work can be summarized as follows:

- We propose a novel and efficien paradigm that enables VLMs to engage in visual reasoning through code generation.

- We construct **Math-VR**, the first large-scale, bilingual dataset and benchmark (178K samples) for Mathematical problems with Visual Reasoning.

- We develop **MatplotCode**, a state-of-the-art image-to-code converter for mathematical figures, and train **CodePlot-CoT** model, a specialized model that achieves up to a 21% performance increase over strong baselines.

## 2 RELATED WORK

**VLMs in Mathematics Reasoning.** Recent progress on multimodal mathematical reasoning largely follows two lines: scaling math-specific multimodal data and improving architectures for visual–text alignment. Representative data-centric efforts include G-LLaVA (Gao et al., 2023a) with a geometry-focused corpus (Geo170K), Math-LLaVA (Shi et al., 2024) with the large-scale MathV360K, and MAVIS (Zhang et al., 2024b) which further optimizes math-specific visual encoding and provides auto-generated CoT-rationales. To evaluate these developments, benchmarks such as MathVista (Lu et al., 2023), MathVerse (Zhang et al., 2024a), Math-Vision (Wang et al., 2024), and MV-Math (Wang et al., 2025b) have emerged, each targeting different aspects of mathematical reasoning in visual contexts. Despite these advances, existing studies have mainly focused on understanding given visual inputs, rather than incorporating visual information into the reasoning chain (plotting auxiliary lines or functions in solutions etc.).

**"Thinking with image" Models.** To overcome the limitations of text-only reasoning, recent research focuses on "Thinking with image," or Visual Chain-of-Thought (VCoT), where models actively retrieve or generate visual aids in reasoning process. Several works (Corbière et al., 2025; Jiang et al., 2025; Zhang et al., 2025; OpenAI, 2025b; Surís et al., 2023; Shao et al., 2024) implement a multimodal chain of thought where they retrieve and crop from the input image and interleave these visual snippets into the reasoning chain to provide more focused context for subsequent VQA steps. Other works (Li et al., 2025a;b; Chern et al., 2025; Pan et al., 2025) focus on building unified models capable of generating interleaved text and image reasoning chain auto-regressively. These approaches discretize or embed images into visual tokens and train a single sequence model that sequentially outputs text and image during decoding. The generated images can serve as visual feedback for navigation tasks like mazes.

**Visual Reasoning Models in Mathematics.** Contemporary visual reasoning models for mathematics predominantly follow two paradigms: interleaved "thinking with image" (Li et al., 2025a; Chen et al., 2025; Wang et al., 2025d) and agent-plus-code tool use (Hu et al., 2024; Gao et al., 2023b; Zhou et al., 2023). Interleaved approaches enable the model to sequentially add auxiliary lines and plot functions. However, the visual actions are weakly controllable, which hampers precise geometric constructions and limits the interpretability of intermediate reasoning steps. The agent-plus-code paradigm treats the model as a planner that create and manipulate input figure by predicting code snippets and call to external tools (Python, CAS/solver libraries, plotting utilities etc.). The executed outputs are again inputted to the model as visual feedback. In contrast to interleaved generation, tool-augmented agents provide precise and verifiable outputs by executing code, yet their performance largely depends on the reliability of the planner. Current planners are often zero-shot and not specifically trained for mathematical reasoning, which makes them susceptible to producing fragile or incorrect tool-use sequences.

## 3 MATH-VR: DATASET AND BENCHMARK FOR MATH VISUAL REASONING

### 3.1 MATHEMATICAL PROBLEMS REQUIRE VISUAL REASONING

Previous mathematical benchmarks, such as Math-500 (vals.ai, 2025b), AIME (vals.ai, 2025a), and HMMT (MathArena, 2025), primarily evaluate models' textual reasoning abilities. More recent efforts like MathVista (Lu et al., 2023) and MATH-Vision (Wang et al., 2024) introduce a multimodal setting by incorporating image-based questions. However, their focus remains largely on visual perception and extracting information from images, and the reasoning processes are still text-only reasoning without introducing visual thoughts.

We argue that visual reasoning in mathematics should be an active process of "reasoning with images", which motivates us to propose a new benchmark, Math-VR. Figure 1(a) illustrates the distinction between existing benchmarks and ours. Previous benchmarks' reasoning can be performed entirely in text, whereas Math-VR necessitates multimodal reasoning with images. For example, the isosceles triangle problem shown in Figure 1(a) requires considering three possible scenarios. Moving beyond naive visual perception, Math-VR demands solvers to conduct reasoning in both text and image domains, such as adding auxiliary lines, to assist with solving math problems.

Table 1: **Key Statistics for Math-VR Benchmark.** We report statistics of our benchmark, including token lengths of questions and solutions, as well as the number and resolution of images.

| Statistics | Number |
|---|---|
| **Question Length (text tokens)** | |
| - Minimum | 9 |
| - Maximum | 602 |
| - Average | 144.23 |
| **Solution Length (text tokens)** | |
| - Minimum | 46 |
| - Maximum | 2753 |
| - Average | 591.14 |
| **# Images in Each Question** | |
| - Maximum number | 4 |
| - Average number | 1.04 |
| - Average resolution | 320x320 |
| **# Images in Each Solution** | |
| - Maximum number | 7 |
| - Average number | 1.24 |
| - Average resolution | 305x305 |

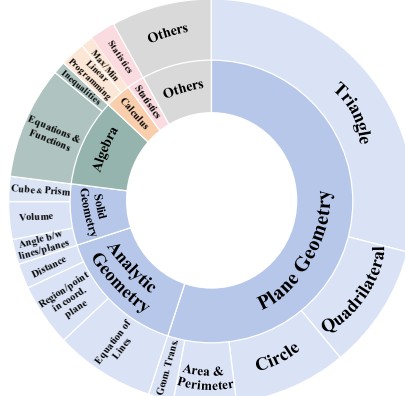

Figure 2: **Distribution of Knowledge Types in Math-VR Benchmark.** Geometry constitutes the majority of problems (77%), with Algebra and Calculus comprising 13%.

### 3.2 DATASET CONSTRUCTION

**Dataset Collection and Filtering.** We begin with collecting 900k secondary-school–level math problems with solutions from public websites, each containing at least one image in explanation (reasoning process) of the solution. Using Qwen2.5-VL-72B, we filter out irrelevant or purely textual images and retain only samples that require mathematical figures for reasoning. Furthermore, we employ GPT-4.1 to convert textual images into readable text and standardize each question into Markdown format. GPT-4.1 further conducts quality checks to discard incomplete or incoherent questions. This process results in **Math-VR dataset**, the first large-scale dataset targeted for visual mathematical reasoning, comprising 178,150 bilingual (English and Chinese) samples.

**Dataset Statistics.** In Math-VR dataset, each sample consists of a question, a reasoning process, and a final answer, with at least one image in the reasoning process. Math-VR encompasses a wide variety of visual reasoning tasks, with 29% text-only and 71% multimodal questions, spanning domains such as Geometry, Algebra, Calculus, and Statistics, where Geometry dominates (81%), and is hierarchically categorized into subdomains and knowledge points (*e.g.*, Triangle, Circle, Quadrilateral, Area, and Perimeter). More details about our dataset collection, categorization, and statistics are presented in Appendix.

### 3.3 BENCHMARK AND EVALUATION

**Benchmark Construction.** To evaluate the visual mathematical reasoning capabilities of different models, we develop the Math-VR benchmark, which consists of 5k bilingual mathematical questions drawn from our dataset. The questions in Math-VR are selected through a careful pipeline designed to ensure a deterministic and reliable evaluation. First, proof-based questions are excluded to avoid the difficulty and bias of assessing the logical validity. Most multiple-choice questions are excluded, since random guessing can yield correct answers by chance. From the remaining questions, a random pool of 3,000 samples was drawn from our dataset and manually reviewed to remove questions that require minimal or trivial visual reasoning.

**Benchmark Statistics and Distribution.** Our Math-VR benchmark is divided into two subsets: the Text subset, containing 2k text-only questions, and the Multimodal subset, comprising 3k questions that are demonstrated with both text and mathematical images. Both subsets require reasoning or imagination in the visual domain to solve the questions. Table 1 and Figure 2 summarize the statistics and distribution of knowledge types of Math-VR benchmark, respectively.

**Evaluation Metrics.** Figure 3 illustrates the evaluation pipeline of Math-VR, where GPT-4.1 (OpenAI, 2025a) is employed as the VLM evaluation tool. We design two metrics: Answer Correctness

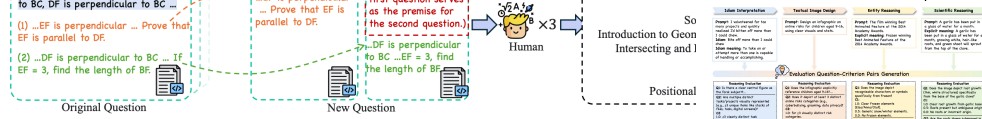

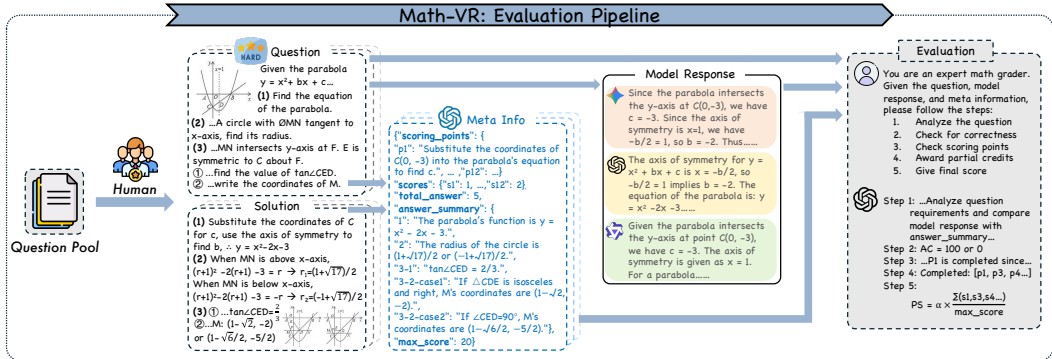

Figure 3: **Math-VR Evaluation Pipeline**. We design a VLM-based framework to comprehensively assess visual reasoning abilities of different models. The evaluation uses two metrics: Answer Correctness (AC), which gives a reliable binary judgment of the final answer, and Process Score (PS), which provides a fine-grained assessment of the solving process.

(AC) and Process Score (PS). Given the free-form nature of the answers (e.g., multiple numbers, ranges, or short text responses), we first use the VLM to analyze the ground-truth solution and generate a comprehensive summary of the final answer for each question. Simultaneously, the VLM is prompted to identify "scoring points" within the solution. These scoring points refer to the critical steps required to solve the problem, such as applying theorems, making necessary deductions, and performing calculations. Each scoring point is assigned a value reflecting its difficulty (e.g., 1 or 2). These extracted answers and scoring points serve as a reference to compare with the model-generated responses during evaluation.

(1) Answer Correctness (AC): To ensure consistent, reproducible, and objective evaluation results, this metric strictly checks whether the model-generated answer matches the ground-truth answer. If the answer is completely correct, it receives a score of 1 for AC; any error or omission results in a score of 0.

(2) Process Score (PS): When solving mathematical questions, even if the final answer is incorrect, the reasoning process may still be meaningful. This metric awards partial credit if the model hits several scoring points in the reasoning process but fails to achieve the completely correct final answer. If a final answer is completely correct (*i.e.*, AC=100), then it automatically receives a PS of 100. Otherwise, PS for a question $q$ is defined as follows:

$$\text{PS}(q) = \alpha \times \frac{\sum_{j=1}^{m} v_j}{\sum_{i=1}^{n} v_i} \times 100, \text{ when the answer is not fully correct} \tag{1}$$

where $\alpha$ represents a discount factor, we take $\alpha = 0.7$. $n$ is the total number of scoring points for the question. $m$ is the number of scoring points hit by the model answer. $v_j$ is the point value of the $j$-th scoring point.

We have conducted a manual review of the VLM summarized information for each sample in our benchmark. For more details about the manual verification and the templates used for prompting GPT-4.1, please refer to the Appendix.

# 4 CODEPLOT-COT PARADIGM: CODE-DRIVEN COT FOR MATHEMATICS VISUAL REASONING

## 4.1 PARADIGM OVERVIEW

Existing VLMs remain constrained when visual aids are required. Most models still rely primarily on text-only chain-of-thought reasoning, which often fails to capture complex visual elements and geometric properties. To address this, recent efforts introduce Visual CoT by directly generating or manipulating images. However, it is difficult for current image generation models to satisfy strict geometric constraints in the mathematical context. These shortcomings constitute fundamental obstacles to accurate visual reasoning in mathematics.

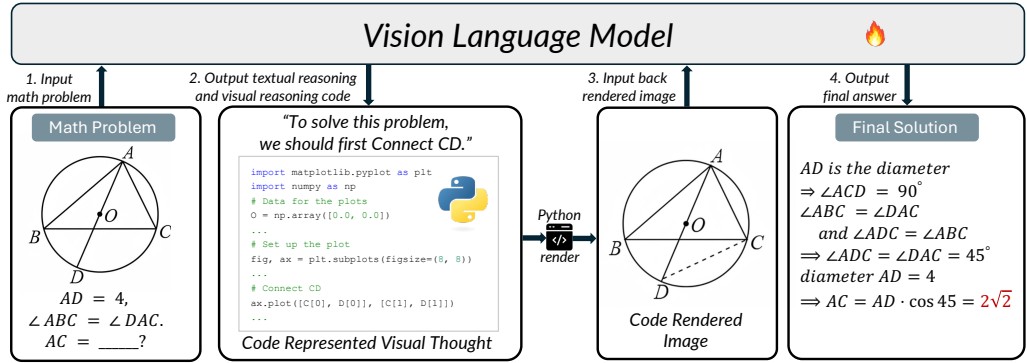

Figure 4: **Illustration of the CodePlot-CoT paradigm for mathematical visual reasoning**. The model interleaves natural language reasoning with code-based visual reasoning. At points in the solution that require visual support, the model generates a sequence of executable plotting code, which is rendered via Python into precise figures and input back into the model. This allows the model to "see" its own visual thought and leverage it for subsequent reasoning, ultimately leading to a more accurate final solution.

To overcome these limitations, we propose CodePlot-CoT, which represents "visual thoughts" as executable plotting code instead of pixel-encoded images. Our key insight is to replace pixel-level visual generation and manipulation with a language modeling problem: instead of "drawing" in visual space, the VLM can "write" code in the text modality where it is inherently proficient. Mathematical figures do not rely on pixel-level fidelity or texture details. Instead, what matters are the structural attributes such as geometric shapes, spatial positions, and angular relations, making executable plotting code a perfect fit for representing such structured geometric information. By releasing the burden of pixel-level distribution modeling, the model can concentrate on the precise geometric information, making it easier to reason with mathematical figures.

Figure 4 presents the CodePlot-CoT paradigm, where text reasoning is interleaved with code-based visual reasoning. The model first generates a reasoning chain in natural language. When visual reasoning step is necessary (such as constructing an auxiliary line), it generates a block of plotting code that represents the required visual information. This code is executed to render an image, which is then input back to the model as the visual reasoning. Such visual thought allows the model to ground its subsequent reasoning in precise, self-generated visual evidence. This fundamentally changes the nature of the model's reasoning, moving it beyond text-only reasoning to a multimodal reasoning chain where thoughts and hypotheses are proposed, tested, and refined in both visual and linguistic domains.

### 4.2 CODE-DRIVEN COT CURATION WITH IMAGE-TO-CODE CONVERTER

A core prerequisite for training CodePlot-CoT is the high-quality data that integrates images, plotting code, and reasoning chains as presented in Figure 4. Such data allows the model to learn how visual thoughts can be faithfully represented as executable code. However, existing mathematical resources rarely provide paired code annotations for visual reasoning, making it difficult to obtain the structured supervision required for code-driven CoT. This motivates the need for a reliable image-to-code mapping that can convert mathematical figures to plotting code. Yet, no fine-grained converter specialized for this domain is publicly available. Even large commercial models (e.g., Gemini-2.5-Pro, GPT-5) are unreliable for zero-shot image-to-code conversion on complex mathematical figures, limiting practical effectiveness. To overcome this bottleneck, we develop **MatplotCode**, a state-of-the-art converter tailored for mathematical figures, which enables scalable creation of code–image pairs and supports the curation of supervised fine-tuning data for CodePlot-CoT.

We leverage the ImgCode-8.6M dataset from MathCoder-VL (Wang et al., 2025a) as the foundation of our experiment. We begin by filtering out images that are not representative of standard mathematical figures, thereby curating a high-quality subset focused on geometry diagrams and function

plots. All code representations in this dataset are in Python. On this curated data, we train Matplot-Code, which demonstrates superior generalization and conversion fidelity. To further enhance the quality of our supervised fine-tuning data, we use MatplotCode to generate multiple Python code representations for each source image and employ GPT-4.1 to select the optimal representation.

### 4.3 TRAINING DETAILS

We leverage Qwen2.5VL-32B as the base model for both MatPlotCode and CodePlot-CoT model. MatPlotCode goes through a two-stage training process: we first align the visual components by training only the vision encoder (ViT) and the MLP projector for one epoch, and then perform full-parameter fine-tuning for two additional epochs. For the CodePlot-CoT model, we initialize its weights from vision-aligned MatPlotCode after Stage 1. We then fully finetune this model on our curated SFT dataset for 5000 steps. During MatPlotCode training, we assign loss to the textual reasoning chain and generated code while no loss is applied to the rendered images in the sequence. More details on our training are presented in the Appendix.

## 5 EXPERIMENTS

### 5.1 HUMAN CORRELATION ANALYSIS

To further validate the reliability of our automatic evaluation pipeline, we conduct a human correlation study. We invite 15 senior undergraduate students majoring in STEM disciplines as our human experts. We then randomly sample 1000 questions from our benchmark and generate a total of 3000 answers from GPT4.1, Gemini-2.5-pro and Claude-Sonnet-4. Each participant is assigned 200 answers and is asked to (i) judge the Answer Correctness (AC) in a correct/incorrect manner and (ii) assign a Process Score (PS) based on the same scoring point in our benchmark. Both scores show strong consistency between human annotations and GPT-4.1 evaluations.

- **Answer Correctness (AC):** We report Cohen's $\kappa = 0.75$ and MCC $= 0.75$. For binary judgments, beyond simple accuracy, these measures account for chance agreement and balance between positive/negative classes.
- **Process Score (PS):** We report Pearson $r = 0.72$ and Spearman $\rho = 0.70$. Pearson captures the linear correlation between human and GPT-4.1 scores, while Spearman focuses on rank-order consistency.

### 5.2 BENCHMARKING EXISTING MODELS AND CODEPLOT-COT ON MATH-VR

For a comprehensive evaluation, we compare our approach against a suite of state-of-the-art LLMs, VLMs, or UMs, including both open-source and closed-source ones on the 2500 English questions in our benchmark, as shown in Table 2. Among closed-source and large open-source models, Gemini-2.5-Pro stands out, achieving the highest overall scores (PS = 80.8, AC = 64.7) on the Math-VR. We observe that "thinking" models which benefit from better-structured textual chains performs better on our benchmark. Among no-thinking models, Nano Banana achieves the highest score, primarily due to its stronger visual reasoning capabilities rather than purely text-based chains. Nevertheless, there remains substantial space for improvement in Answer Correctness across closed-source models. Even the strongest performer, Gemini-2.5-Pro, still fails on approximately one-third of the benchmark problems. This gap highlights that current advances in chain length and scale alone are insufficient, and points toward future research directions in developing more faithful and controllable visual reasoning mechanisms.

The limitations are more significant in open-source models under 100B, most of which remain constrained to pure text reasoning and consequently exhibit low answer correctness on Math-VR benchmark (e.g., Gemma, Qwen2.5-VL-72B). Attempts to "think with images" by directly generating pixels (e.g., Bagel-Zebra-CoT) provide only modest gains due to poor controllability and low geometric precision due to model scale. In contrast, our code-driven approach yields substantial improvements: CodePlot-CoT surpasses its 32B base VLM by up to 21% and largely outperforms the Qwen2.5-VL-72B across all metrics, demonstrating that structured, verifiable visual reasoning is more decisive than model size or longer textual chains.

Table 2: **Math-VR evaluation results.** This table compares the model performances on our benchmark. The second column specifies the model size by its parameter count. For models that support an internal reasoning process, ✓in the "Thinking" column indicate this mode is turned on during evaluation. Model Types: VLM: Vision Language Model, LLM: Large Language Model, UM: Unified Model. Metrics: PS: Process Score, AC: Answer Correctness. Blue and Yellow highlight the top score in each model group. **Bold** signifies the highest score across all models.

| Model | # Params | Type | Think | Text | | Multimodal | | Overall | |
|---|---|---|---|---|---|---|---|---|---|
| | | | | PS | AC | PS | AC | PS | AC |
| *Closed-source Models and Open-source Models over 100B* | | | | | | | | | |
| GPT-4o(2024) | - | VLM | × | 34.6 | 5.7 | 27.6 | 3.4 | 30.4 | 4.3 |
| GPT-4.1-nano(2025a) | - | VLM | × | 45.9 | 13.1 | 33.6 | 6.4 | 38.5 | 9.1 |
| GPT-4.1-mini(2025a) | - | VLM | × | 62.0 | 33.3 | 58.6 | 33.3 | 60.0 | 33.3 |
| GPT-4.1(2025a) | - | VLM | × | 56.5 | 26.6 | 52.2 | 25.6 | 53.9 | 26.0 |
| GPT-o3(2025b) | - | VLM | ✓ | 72.9 | 52.9 | 78.6 | 63.7 | 76.4 | 59.3 |
| Gemini-2.0-Flash(2024) | - | VLM | × | 56.1 | 24.1 | 47.0 | 18.3 | 50.7 | 20.6 |
| Gemini-2.5-Flash(2025) | - | VLM | × | 70.9 | 44.6 | 75.5 | 57.5 | 73.7 | 52.3 |
| Gemini-2.5-Flash(2025) | - | VLM | ✓ | 77.5 | 57.0 | 79.0 | 62.9 | 78.4 | 60.5 |
| Gemini-2.5-Pro(2025) | - | VLM | ✓ | **77.9** | **58.7** | **82.8** | **68.7** | **80.8** | **64.7** |
| Nano Banana(2025) | - | UM | × | 72.3 | 49.1 | 74.7 | 56.3 | 73.8 | 53.4 |
| Seed-1.6-Thinking(2025) | - | VLM | ✓ | 73.0 | 53.0 | 76.6 | 62.0 | 75.2 | 58.4 |
| Claude-Sonnet-4(2025) | - | VLM | × | 60.9 | 31.5 | 53.4 | 25.8 | 56.4 | 28.1 |
| GLM-4.5V(2025) | 108B | VLM | ✓ | 70.5 | 48.0 | 69.1 | 50.6 | 69.7 | 49.6 |
| Deepseek-R1(2025) | 671B | LLM | ✓ | 69.9 | 49.5 | - | - | - | - |
| *Open-source Models under 100B* | | | | | | | | | |
| Bagel(2025) | 7B | UM | × | 32.9 | 8.5 | 24.0 | 7.0 | 27.6 | 7.6 |
| Bagel-Zebra-CoT(2025a) | 7B | UM | × | 41.5 | 13.9 | 29.1 | 7.6 | 34.1 | 10.1 |
| Keye-VL-1.5(2025) | 8B | VLM | × | 44.4 | 20.2 | 34.0 | 15.4 | 38.2 | 17.3 |
| InternVL-3.5-8B(2025c) | 8B | VLM | × | 35.6 | 9.2 | 28.6 | 7.0 | 31.4 | 7.9 |
| Gemma3(2025) | 27B | VLM | × | 50.8 | 19.2 | 40.8 | 14.1 | 44.8 | 16.1 |
| Qwen-2.5-VL-3B(2025) | 3B | VLM | × | 33.4 | 7.9 | 23.6 | 3.6 | 27.5 | 5.3 |
| Qwen-2.5-VL-7B(2025) | 7B | VLM | × | 18.0 | 4.5 | 11.0 | 2.0 | 13.8 | 3.0 |
| Qwen-2.5-VL-32B(2025) | 32B | VLM | × | 36.9 | 10.6 | 31.5 | 9.6 | 33.7 | 10.0 |
| Qwen-2.5-VL-72B(2025) | 72B | VLM | × | 44.6 | 15.3 | 38.2 | 12.7 | 40.8 | 13.7 |
| CodePlot-CoT | 32B | VLM | × | 53.8 | 31.6 | 42.4 | 15.8 | 47.0 | 22.1 |
| Δ Over Base Model | | | | +16.9 | +21.0 | +10.9 | +6.2 | +13.3 | +12.1 |

## 5.3 IMAGE-CODE CONVERTER EVALUATION

We evaluate MatplotCode against FigCodifier-8B, GPT-o3 and Gemini-2.5-pro (thinking budget maximum). We randomly sample 1000 images from our dataset and task these models to convert it to matplotlib code. We assess two aspects: (i) Execution Success Rate, i.e., the probability that the generated code runs without errors. (ii) Reconstruction Fidelity, judged by GPT-4.1 via a standardized prompt to decide which reconstruction is most similar to the original image. The full evaluation prompt is provided in the Appendix.

Both MatplotCode and the FigCodifier-8B achieve a 100% execution success rate. In contrast, GPT-o3 reach 79.6%, while Gemini-2.5-Pro achieve 86.2%, indicating a higher likelihood of producing invalid or incomplete code. For reconstruction fidelity, judged by GPT-4.1, MatplotCode is preferred in 554 out of 1,000 cases, compared to 190 for FigCodifier-8B, 49 for GPT-o3, and 207 for Gemini-2.5-Pro. These results demonstrate that our converter not only guarantees reliable code prediction, but also produces reconstructions that are consistently closer to the original figures. Importantly, the failures of both closed-source large models and open-source expert models further underscore the necessity of developing a new converter tailored to this task.

Table 3: Comparing our code-driven paradigm against text-only reasoning. Qwen-2.5VL-3B-Text-Tune is fine-tuned on text-only CoT, while CodePlot-CoT-3B is trained with our paradigm. The results demonstrate the significant performance gain from enabling code-based visual reasoning.

| Ablation Setting | Text | | Multimodal | | Overall | |
|---|---|---|---|---|---|---|
| | PS | AC | PS | AC | PS | AC |
| Qwen-2.5VL-3B | 33.4 | 7.9 | 23.6 | 3.6 | 27.5 | 5.3 |
| Qwen-2.5VL-3B-Text-Tune | 34.3 | 12.7 | 27.4 | 5.8 | 30.1 | 8.4 |
| CodePlot-CoT-3B | **35.5** | **13.6** | **29.3** | **7.4** | **31.8** | **9.9** |

Table 4: Comparing our code-driven paradigm against direct image generation VCoT. Bagel-Thinking-with-image is fine-tuned to generate direct image outputs in reasoning steps, while CodePlot-CoT-Bagel uses our code-generation paradigm. This validates the efficacy of our paradigm.

| Ablation Setting | Text | | Multimodal | | Overall | |
|---|---|---|---|---|---|---|
| | PS | AC | PS | AC | PS | AC |
| Bagel | 32.9 | 8.5 | 24.0 | 7.0 | 27.6 | 7.6 |
| Bagel-Thinking-with-image | 40.3 | 10.1 | 28.4 | 8.3 | 33.2 | 9.0 |
| CodePlot-CoT-Bagel | **43.1** | **11.9** | **31.1** | **10.2** | **35.9** | **10.9** |

## 5.4 ANALYSIS ON INFERENCE COST

In this section, we analyze the inference cost of our code-driven visual reasoning paradigm. On the 2,500 test problems, our model generates an average of 820.9 tokens per image and 3,416 rendered images in total, or 1.37 images per problem in average. From the perspective of token usage, this cost is lower than many autoregressive "thinking with image" models, which typically consume 1,024 or even up to 4,096 tokens per image. Moreover, each image can be rendered locally in less than one second, making it negligible when evaluating overall computational efficiency. We further measure the total number of output tokens. On average, our model produces 567.2 text tokens, and together with the code tokens for rendered images, the overall output is 1,691.8 tokens per problem. In comparison, Qwen2.5-VL-32B generates substantially more content, averaging 3,847.3 tokens. This significant reduction in output length highlights the efficiency of our code-driven visual reasoning paradigm on Math-VR.

## 5.5 ABLATION STUDIES

To better understand the contribution of each design choice in our framework, we conduct two sets of ablation experiments.

**Text-only vs. Code-driven Visual Reasoning.** We first compare our code-driven paradigm against text-only reasoning using Qwen-2.5VL-3B as the base model. To establish a text-only fine-tuned baseline, we remove all images from the solutions in our dataset and fine-tune the model exclusively on textual reasoning, resulting in Qwen-2.5VL-3B-Text-Tune. This model shows only marginal improvement over the vanilla baseline because it remains fundamentally constrained by its inability to incorporate visual information. In contrast, our CodePlot-CoT-3B achieves substantial gains, clearly demonstrating the advantage of introducing executable code for visual reasoning.

**Code-driven vs. Direct Image Generation.** We further investigate whether code-based visual reasoning is more effective than direct image generation in reasoning. Since Qwen2.5-VL does not support image generation, we perform this comparison on the unified model Bagel. The Bagel-Thinking-with-image is fine-tuned to directly produce interleaved text–image outputs on our dataset. Although this approach provides some improvements, it underperforms our CodePlot-CoT-Bagel, which leverages structured executable code. These results validate that code-driven reasoning offers a more precise and controllable representation of visual thoughts than direct pixel-level generation.

## 6 CONCLUSION

In this work, we introduce CodePlot-CoT, a code-driven chain-of-thought paradigm that enables VLMs to "think with images" in mathematical reasoning. By representing visual reasoning as executable snippets of plotting code, our approach circumvents the limitations of pixel-level image generation, achieving precise and controllable visual thought. To achieve this paradigm, we construct Math-VR, the first large-scale bilingual dataset benchmark for mathematical visual reasoning, and develop MatplotCode, a high-fidelity image-to-code converter. Extensive experiments demonstrate that our model consistently outperforms baseline models with improvements of up to 21%. We have used GPT-5 to help refine grammar in this paper.

# 7 ETHICS AND REPRODUCIBILITY STATEMENT

All data used in this study are collected from publicly available websites, ensuring that no private or sensitive information is involved in the dataset construction. The details on dataset, benchmark construction and evaluation are presented in Section 3.2 and Section 3.3. Training and implementation details are described in Section 4.3. Additional information, including evaluation templates, manual verification processes, and further dataset construction details, is provided in the Appendix. These resources are made available to facilitate transparent assessment and to support reproducibility of our results.

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
