# OpenReview forum: "CodePlot-CoT: Mathematical Visual Reasoning by Thinking with Code-Driven Images"
_ICLR.cc/2026/Conference — ICLR 2026 Conference Withdrawn Submission_

### Official Review · Reviewer_9xdz · 2025-10-26

**Soundness:** 2
**Presentation:** 2
**Contribution:** 2
**Rating:** 4
**Confidence:** 4

**Summary:**

This paper presents an interesting and well-motivated approach to code-driven visual reasoning in mathematics, but its limited generalization scope and insufficient evaluation details weaken its overall impact.

**Strengths:**

The paper introduces Math-VR, a new benchmark that requires multimodal reasoning involving both textual and visual understanding. The example in Figure 1 effectively illustrates how visual information plays an integral role in the reasoning process. Existing benchmarks have largely overlooked this kind of testing cases.

The idea of converting between code and plots to construct and refine mathematical figures is novel and appealing. This design makes the approach particularly suitable for reasoning tasks involving geometric or visually grounded mathematical problems.

**Weaknesses:**

Although the proposed paradigm of code generation for multimodal reasoning is interesting, its applicability appears quite narrow. The approach mainly fits geometric or mathematical reasoning tasks and may not generalize well to broader natural image reasoning scenarios. This limits the overall utility and impact of the method.

While Figure 1(a) presents an engaging example, Section 3.2 lacks sufficient detail on how the authors ensure that the collected data samples necessitate multimodal reasoning with images. The section mainly describes general filtering steps (e.g., removing irrelevant or purely textual samples) without explaining how the desired reasoning characteristics were verified.

In Section 3.3, it remains unclear how the authors confirm that the selected benchmark samples genuinely require reasoning or imagination in the visual domain to solve. Further clarification or examples would help strengthen the validity of the dataset design.

The evaluation pipeline appears to rely on proprietary models such as GPT-4.1 to serve as judges. If that is indeed the case, it introduces a considerable evaluation cost. It would be helpful if the authors could discuss whether alternative, more accessible models were tested as evaluators and how consistent the results were.

The experiments are conducted solely on the proposed Math-VR dataset, which limits the evaluation scope. Since both training and testing are performed within the same domain, it is difficult to assess the generalization ability of the proposed CodePlot-CoT model. The authors are encouraged to include evaluations on other multimodal reasoning benchmarks to better demonstrate the robustness of their approach.

According to Table 3, the performance gap between Qwen-2.5VL-3B-Text-Tune and CodePlot-CoT-3B is quite small, especially in the text category where the difference is less than one point. This raises the question of whether the proposed code-generation mechanism truly contributes to performance on text-based questions. Some further analysis or ablation discussion would help clarify this point.

**Questions:**

Please see weaknesses.

---

### Official Review · Reviewer_EWzn · 2025-10-27

**Soundness:** 3
**Presentation:** 3
**Contribution:** 3
**Rating:** 4
**Confidence:** 4

**Summary:**

This paper proposes CodePlot-CoT, a new framework that lets vision-language models solve math problems by “thinking with code.” Instead of directly generating images, the model writes executable plotting code, renders it into a figure, and uses that image for further reasoning. To support this, this paper build Math-VR, the first large-scale bilingual dataset (178K samples) for mathematical visual reasoning, and design MatplotCode, a high-quality image-to-code converter. Experiments show that CodePlot-CoT improves performance by up to 21%over strong baselines, proving that code-based visual reasoning is more precise and controllable than text-only or pixel-based methods.

**Strengths:**

1. Turning visual reasoning into a code generation task is creative and practical, addressing the precision problem of image-based reasoning.

2. Math-VR is large, bilingual, and fills an important gap in visual math reasoning research.

3. The MatplotCode converter ensures high-fidelity code–image alignment for training and evaluation.

**Weaknesses:**

1. Although the code-driven reasoning paradigm shows clear advantages on Math-VR, the evaluation is limited to a benchmark that mainly emphasizes visual reasoning rather than visual perception. Current results do not demonstrate how the proposed CodePlot-CoT performs on perception-centric benchmarks such as MathVista, MATH-Vision, or MathVerse. Evaluating on these benchmarks would better reveal whether the proposed method can generalize beyond “reasoning with generated figures” to “understanding externally provided visual inputs,” thereby validating its applicability to a wider multimodal reasoning spectrum.

2. The paper conceptually discusses the difference between code-driven CoT and agent-plus-code approaches—highlighting that CodePlot-CoT is end-to-end trained rather than relying on zero-shot planning—but this distinction remains mainly theoretical. The experiments lack a direct, quantitative comparison with representative agent-style systems (e.g., PAL, ViperGPT, or MathCoder-VL).
Without such a head-to-head analysis, it is difficult to assess whether the observed gains come from the structured code-based representation itself or simply from better supervision.

3. The current training pipeline is purely supervised fine-tuning (SFT), where the model learns to produce both textual reasoning and plotting code. However, the model is not explicitly trained to decide when to invoke code generation, how complex the generated figure should be, or how many iterations of visualization are optimal.
As a result, the model tends to “always or frequently write code,” which can lead to redundant plotting and unnecessary computation. This limits both efficiency and interpretability.

4. The dataset and experiments focus predominantly on planar geometry, which constitutes over 70% of the Math-VR corpus. While this focus highlights the advantages of structured visual reasoning, it also raises concerns about generalization to other mathematical domains such as 3D or analytic geometry, functional plots, or composite graph reasoning.

**Questions:**

same as weakness

---

### Official Review · Reviewer_MumB · 2025-10-28

**Soundness:** 2
**Presentation:** 2
**Contribution:** 2
**Rating:** 2
**Confidence:** 5

**Summary:**

This paper introduces CodePlot-CoT, a code-driven chain-of-thought framework designed to enhance multimodal mathematical reasoning by enabling models to “think with images.” They propose integrating code generation and execution into the reasoning loop, allowing the model to render auxiliary figures dynamically. They also construct Math-VR, a large-scale bilingual benchmark (178K samples) focusing on visual reasoning in mathematics, and develop a specialized image-to-code converter for converting complex mathematical diagrams into executable plotting code.

**Strengths:**

The paper delivers both a large-scale dataset (Math-VR) and a new reasoning paradigm (CodePlot-CoT), which together form a strong foundation for future research on multimodal mathematical reasoning.

The study addresses a real limitation of current VLMs and LLMs—poor handling of visual reasoning in mathematics, particularly for auxiliary constructions and geometry.

The paper is clearly written and structured, with consistent problem formulation and experimental design.

**Weaknesses:**

The most substantial contribution appears to be the Math-VR benchmark itself, rather than a fundamentally new reasoning paradigm. The idea of “drawing as reasoning” has already been explored in works such as Visual Sketchpad: Sketching as a Visual Chain of Thought for Multimodal Language Models, as well as in workflow-based and trainable agentic frameworks (e.g., rStar2-Agent: Agentic Reasoning Technical Report), which also employ Python or MCP as reasoning tools. The current approach provides limited methodological novelty beyond adapting these ideas to mathematical diagrams.

This method uses Python primarily as a drawing tool but does not explain how to establish visual reward supervision. The model receives rewards only for final reasoning accuracy, without explicit supervision for visual correctness or perceptual quality of the generated diagrams. This weakens the learning signal for visual grounding and does not effectively address the diagram hallucination issues commonly observed in current VLMs. Moreover, training solely on the proposed dataset risks overfitting and may not truly enhance the model’s visual perception ability for broader AI4Math or AI4Science tasks. The authors should include more comprehensive comparisons with existing methods on public benchmarks to better demonstrate generalization and robustness.

The model adopts the same “thinking with images” pipeline used for natural image reasoning, with no algorithmic modification to accommodate the unique characteristics of abstract diagrams—aside from the construction of the tracing set. As is well known, minor geometric edits (e.g., adding auxiliary lines) often produce subtle visual changes that standard vision encoders (e.g., DINO, GLIP), which are trained on natural images, typically fail to capture effectively.

The Python plotting code represents only primitive geometric elements (points, lines, coordinates) necessary for rendering, but lacks symbolic abstraction (e.g., shapes, relations, symmetries). This limits reasoning depth, as mathematical diagram understanding depends on higher-level symbolic relations rather than pixel or coordinate features.

**Questions:**

The dataset generation pipeline depends on Qwen2.5-VL-72B and GPT-4.1 for parsing and standardization. However, both models exhibit limitations in accurately interpreting diagrams and mathematical text, raising questions about annotation fidelity and noise control in Math-VR.

The method and dataset focus narrowly on geometric problem solving. It remains unclear how the proposed framework generalizes to broader mathematical or visual reasoning domains, such as algebraic graph reasoning, physics diagrams, or function plots.

**Details Of Ethics Concerns:**

No ethics concerns.

---

### Official Review · Reviewer_kPp4 · 2025-10-29

**Soundness:** 2
**Presentation:** 3
**Contribution:** 3
**Rating:** 4
**Confidence:** 4

**Summary:**

This paper presents a novel and compelling paradigm, CodePlot-CoT, for addressing the challenging problem of visual reasoning in mathematics. The introduction of the Math-VR dataset and the code-driven approach are significant contributions. The experimental design is generally comprehensive, with extensive benchmarking.

**Strengths:**

Addresses a timely and compelling research topic by tackling the critical challenge of precise visual reasoning in mathematical problem-solving through a code-driven approach. Provides a large-scale, bilingual dataset (Math-VR, 178K samples) that establishes a comprehensive benchmark for mathematical visual reasoning, offering substantial potential to benefit related research.

**Weaknesses:**

1. The α=0.7 discount for incorrect answers is arbitrary and lacks justification, making PS improvements uninterpretable.
2. Awarding a perfect PS=100 for any correct answer, regardless of reasoning quality, invalidates the metric's purpose of evaluating the reasoning process itself.
3. The entire PS relies on "scoring points" extracted by GPT-4.1. No inter-rater reliability or consistency analysis is provided for this highly subjective core process, making the benchmark non-reproducible and potentially biased.
4. The paper references an Appendix (e.g...,line 424) that was not provided for review. This prevents verification of crucial methodological details and violates basic standards of reproducibility.

**Questions:**

As above

---

### Note · Authors · 2025-11-14

I have read and agree with the venue's withdrawal policy on behalf of myself and my co-authors.